# STUNT: Few-shot Tabular Learning with Self-generated Tasks from Unlabeled Tables

**Jaehyun Nam**[1]  **Jihoon Tack**[1]  **Kyungmin Lee**[1]  **Hankook Lee**[2*]  **Jinwoo Shin**[1]
[1]Korea Advanced Institute of Science and Technology (KAIST)  [2]LG AI Research
{jaehyun.nam, jihoontack, kyungmnlee, jinwoos}@kaist.ac.kr
hankook.lee@lgresearch.ai

## Abstract

Learning with few labeled tabular samples is often an essential requirement for industrial machine learning applications as varieties of tabular data suffer from high annotation costs or have difficulties in collecting new samples for novel tasks. Despite the utter importance, such a problem is quite under-explored in the field of tabular learning, and existing few-shot learning schemes from other domains are not straightforward to apply, mainly due to the heterogeneous characteristics of tabular data. In this paper, we propose a simple yet effective framework for few-shot semi-supervised tabular learning, coined *Self-generated Tasks from UNlabeled Tables (STUNT)*. Our key idea is to self-generate diverse few-shot tasks by treating randomly chosen columns as a target label. We then employ a meta-learning scheme to learn generalizable knowledge with the constructed tasks. Moreover, we introduce an unsupervised validation scheme for hyperparameter search (and early stopping) by generating a pseudo-validation set using STUNT from unlabeled data. Our experimental results demonstrate that our simple framework brings significant performance gain under various tabular few-shot learning benchmarks, compared to prior semi- and self-supervised baselines. Code is available at https://github.com/jaehyun513/STUNT.

## 1 Introduction

Learning with few labeled samples is often an essential ingredient of machine learning applications for practical deployment. However, while various few-shot learning schemes have been actively developed over several domains, including images (Chen et al., 2019) and languages (Min et al., 2022), such research has been under-explored in the tabular domain despite its practical importance in industries (Guo et al., 2017; Zhang et al., 2020; Ulmer et al., 2020). In particular, few-shot tabular learning is a crucial application as varieties of tabular datasets (i) suffer from high labeling costs, e.g., the credit risk in financial datasets (Clements et al., 2020), and (ii) even show difficulties in collecting new samples for novel tasks, e.g., a patient with a rare or new disease (Peplow, 2016) such as an early infected patient of COVID-19 (Zhou et al., 2020).

To tackle such limited label issues, a common consensus across various domains is to utilize unlabeled datasets for learning a generalizable and transferable representation, e.g., images (Chen et al., 2020a) and languages (Radford et al., 2019). Especially, prior works have shown that representations learned with self-supervised learning are notably effective when fine-tuned or jointly learned with few labeled samples (Tian et al., 2020; Perez et al., 2021; Lee et al., 2021b; Lee & Shin, 2022). However, contrary to the conventional belief, we find this may not hold for tabular domains. For instance, recent state-of-the-art self-supervised tabular learning methods (Yoon et al., 2020; Ucar et al., 2021) do not bring meaningful performance gains over even a simple k-nearest neighbor (kNN) classifier for few-shot tabular learning in our experiments (see Table 1 for more details). We hypothesize that this is because the gap between trained self-supervised tasks and the applied few-shot task is large due to the heterogeneous characteristics of tabular data.

Instead, we ask whether one can utilize the power of meta-learning to reduce the gap via fast adaption to unseen few-shot tasks; meta-learning is indeed one of the most effective few-shot learning

---

*Work done at KAIST.

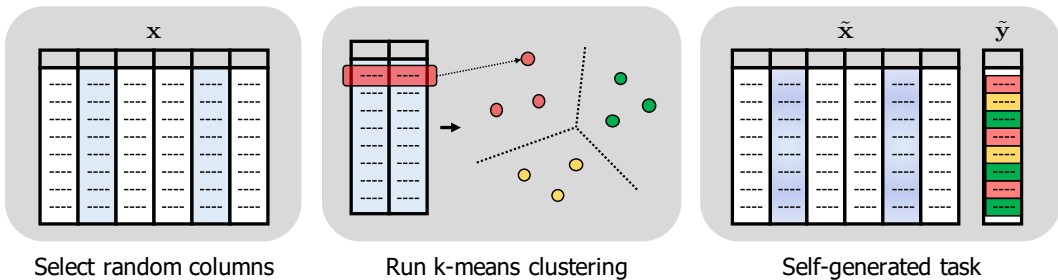

Figure 1: An overview of the proposed *Self-generated Tasks from UNlabeled Tables (STUNT)*: we generate the task label by running a k-means clustering over the randomly selected column features of the table, then perturb the selected columns to prevent from generating a trivial task.

strategies across domains (Finn et al., 2017; Gu et al., 2018; Xie et al., 2018). We draw inspiration from the recent success in unsupervised meta-learning literature, which meta-learns over the self-generated tasks from unlabeled data to train an effective few-shot learner (Khodadadeh et al., 2019; Lee et al., 2021a). It turns out that such an approach is quite a promising direction for few-shot tabular learning: a recent unsupervised meta-learning scheme (Hsu et al., 2018) outperforms the self-supervised tabular learning methods in few-shot tabular classification in our experiments (see Table 1). In this paper, we suggest to further exploit the benefits of unsupervised meta-learning into few-shot tabular learning by generating more diverse and effective tasks compared to the prior works using the distinct characteristic of the tabular dataset's column feature.

**Contribution.** We propose a simple yet effective framework for few-shot semi-supervised tabular learning, coined *Self-generated Tasks from UNlabeled Tables* (STUNT); see the overview in Figure 1. Our key idea is to generate a diverse set of tasks from the unlabeled tabular data by treating the table's column feature as a useful target, e.g., the 'blood sugar' value can be used as a substituted label for 'diabetes'. Specifically, we generate pseudo-labels of the given unlabeled input by running a k-means clustering on randomly chosen subsets of columns. Moreover, to prevent generating a trivial task (as the task label can be directly inferred by the input columns), we randomly replace the chosen column features with a random value sampled from the columns' respective empirical marginal distributions. We then apply a meta-learning scheme, i.e., Prototypical Network (Snell et al., 2017), to learn generalizable knowledge with the self-generated tasks.

We also find that the major difficulty of the proposed meta-learning with unlabeled tabular datasets is the absence of a labeled validation set; the training is quite sensitive to hyperparameter selection or even suffers from overfitting. To this end, we propose an unsupervised validation scheme by utilizing STUNT to the unlabeled set. We find that the proposed technique is highly effective for hyperparameter searching (and early stopping), where the accuracy of the pseudo-validation set and the test set show a high correlation.

We verify the effectiveness of STUNT through extensive evaluations on various datasets in the OpenML-CC18 benchmark (Vanschoren et al., 2014; Bischl et al., 2021). Overall, our experimental results demonstrate that STUNT consistently and significantly outperforms the prior methods, including unsupervised meta-learning (Hsu et al., 2018), semi- and self-supervised learning schemes (Tarvainen & Valpola, 2017; Yoon et al., 2020; Ucar et al., 2021) under few-shot semi-supervised learning scenarios. In particular, our method improves the average test accuracy from 59.89%→63.88% for 1-shot and from 72.19%→74.77% for 5-shot, compared to the best baseline. Furthermore, we show that STUNT is effective for multi-task learning scenarios where it can adapt to new tasks without retraining the network.

## 2 RELATED WORK

**Learning with few labeled samples.** To learn an effective representation with few labeled samples, prior works suggest leveraging the unlabeled samples. Such works can be roughly categorized as (i) semi-supervised (Kim et al., 2020; Assran et al., 2021) and (ii) self-supervised (Chen et al., 2020a;b)

approaches. For semi-supervised learning approaches, a common way is to produce a pseudo-label for each unlabeled data by using the model's prediction and then train the model with the corresponding pseudo-label (Lee, 2013). More advanced schemes utilize the momentum network (Laine & Aila, 2017; Tarvainen & Valpola, 2017) and consistency regularization with data augmentations (Berthelot et al., 2019; Sohn et al., 2020) to generate a better pseudo-label. On the other hand, self-supervised learning schemes aim to pre-train the representation by using domain-specific inductive biases (e.g., the spatial relationship of image augmentations), then fine-tune or adapt with a few labeled samples (Tian et al., 2020; Perez et al., 2021). In particular, previous studies have shown the effectiveness of self-supervised learning in the transductive setting compared to the few-shot methods (Chen et al., 2021). While recent works on both approaches heavily rely on augmentation schemes, it is unclear how to extend such methods to the tabular domain due to the heterogeneous characteristics of tabular datasets, i.e., there is no clear consensus on which augmentation is globally useful for the tabular dataset. Instead, we train the unlabeled dataset with an unsupervised meta-learning framework that does not rely on the effect of augmentation.

**Learning with unlabeled tabular data.** Various attempts have been made to train a generalizable representation for tabular datasets using unlabeled samples. Yoon et al. (2020) is the first to target self-supervised learning on the tabular dataset by corrupting random features and then predicting the corrupted location (i.e., row and columns). As a follow-up, Bahri et al. (2022) directly adopts contrastive learning frameworks (Chen et al., 2020a) by corrupting randomly selected features to create positive views, and Ucar et al. (2021) shows that using effective three pretext task losses (i.e., reconstruction loss, contrastive loss, and distance loss) can achieve the state-of-the-art performance on the linear evaluation task (training a linear classifier with a labeled dataset upon the learned representation). However, while prior works have shown their effectiveness mainly on linear evaluation, we find that these methods may not be effective for few-shot learning scenarios (see Table 1). In this regard, we suggest utilizing the power of meta-learning for training an effective few-shot learner by proposing an unsupervised meta-learning framework for tabular data. Moreover, while some works have proposed a self-supervised learning framework (Somepalli et al., 2021; Majmundar et al., 2022) based on Transformer architectures (Vaswani et al., 2017), we believe a new approach is needed for few-shot learning (given the observations of prior self-supervised learning works in Table 1) and introduce an architecture-agnostic method that can be used for a wide range of applications.

**Unsupervised meta-learning.** Meta-learning, i.e., learning to learn by extracting common knowledge over a task distribution, has emerged as a popular paradigm for enabling systems to adapt to new tasks in a sample-efficient way (Vinyals et al., 2016; Finn et al., 2017; Snell et al., 2017). Recently, several works have suggested unsupervised meta-learning schemes that self-generate a labeled task from the unlabeled dataset, as various few-shot learning applications suffer from high annotation costs. To self-generate the tasks, CACTUs (Hsu et al., 2018) run a clustering algorithm on a representation trained with self-supervised learning, Ye et al. (2022) and UMTRA (Khodadadeh et al., 2019) assumes the augmented sample as the same pseudo-class (Ye et al. (2022) also introduces effective strategies for unsupervised meta-learning: sufficient episodic sampling, hard mixed supports, and task specific projection head), and Meta-GMVAE (Lee et al., 2021a) utilize a variational autoencoder (VAE; Kingma & Welling (2014)) for clustering. However, despite the effectiveness of prior works on image datasets, we find that applying each method to the tabular domain is highly non-trivial; they assume high-quality self-supervised representation, data augmentations, or sophisticated generative models like VAE which are quite cumbersome to have in the tabular domain, e.g., see the work by Xu et al. (2019). Nevertheless, we have tried the prior methods for tabular learning using various techniques, e.g., several augmentations such as noise and permutation, but only CACTUs has shown its effectiveness; we find that Meta-GMVAE underperforms the baseline with the lowest performance that we consider. In this paper, we propose a new unsupervised meta-learning that is specialized for tabular datasets.

## 3   STUNT: SELF-GENERATED TASKS FROM UNLABELED TABLES

In this section, we develop an effective few-shot tabular learning framework that utilizes the power of meta-learning in an unsupervised manner. In a nutshell, our framework meta-learns over the self-generated tasks from the unlabeled tabular dataset, then adapt the network to classify the few-shot labeled dataset. We first briefly describe our problem setup (Section 3.1), and then the core component, coined *Self-generated tasks from UNlabeled Tables (STUNT)*, which generates effective

and diverse tasks from the unlabeled tabular data (Section 3.2). Moreover, we introduce a pseudo-validation scheme with STUNT, where one can tune hyperparameters (and apply early stopping) even without a labeled validation set on few-shot scenarios (Section 3.3).

## 3.1 PROBLEM SETUP: FEW-SHOT SEMI-SUPERVISED LEARNING

We first describe the problem setup of our interest, the few-shot semi-supervised learning for classification. Formally, our goal is to train a neural network classifier $f_\theta : \mathcal{X} \to \mathcal{Y}$ parameterized by $\theta$ where $\mathcal{X} \subseteq \mathbb{R}^d$ and $\mathcal{Y} = \{0,1\}^C$ are input and label spaces with $C$ classes, respectively, and assume that we have a labeled dataset $\mathcal{D}_l = \{\mathbf{x}_{l,i}, \mathbf{y}_{l,i}\}_{i=1}^{N_l} \subseteq \mathcal{X} \times \mathcal{Y}$ and an unlabeled dataset $\mathcal{D}_u = \{\mathbf{x}_{u,i}\}_{i=1}^{N_u} \subseteq \mathcal{X}$ for training the classifier $f_\theta$. Note that all the data points are sampled from a certain data-generating distribution $p(\mathbf{x}, \mathbf{y})$ in an *i.i.d.* manner. We also assume that the cardinality of the given labeled set is very small, e.g., one sample per class, while we have a sufficient amount of the unlabeled dataset, i.e., $N_u \gg N_l$.

## 3.2 UNSUPERVISED META-LEARNING WITH STUNT

We now describe the core algorithm we propose, STUNT. To obtain a good classifier $f_\theta$ under the proposed setup, we suggest using an unsupervised meta-learning method, which (i) self-generates diverse tasks $\{\mathcal{T}_1, \mathcal{T}_2, \ldots\}$ from the unlabeled dataset $\mathcal{D}_u$ where each $\mathcal{T}_i$ contains few samples with pseudo-labels; (ii) meta-learns $f_\theta$ to generalize across the tasks; and, (iii) adapts the classifier $f_\theta$ using the labeled dataset $\mathcal{D}_l$. Algorithm 1 in Appendix D provides the detailed training process.

**Task generation from unlabeled tables.** Our key idea is to generate a diverse set of tasks from the unlabeled data by treating a column feature of tabular data as a useful pseudo-label. Intuitively speaking, as any label type can be considered as a tabular column due to the heterogeneous characteristic of the tabular data (i.e., each column has a distinct feature), one can also rethink any column feature as a task label. In particular, since there exist some columns that have a high correlation with the original label, the new task constructed with such a column feature is highly similar to the original classi-

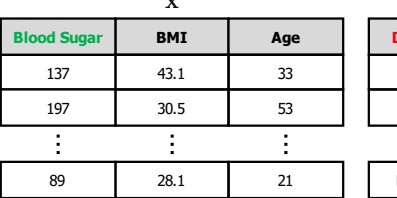

Figure 2: Example data from the diabetes dataset (Bischl et al., 2021). The red column indicates the original target, and the green column indicates the possible alternative label.

fication task, e.g., the original task of predicting 'Diabetes' through 'BMI' and 'Age' is similar to a new task that predicts 'Blood Sugar' by 'BMI' and 'Age' (see Figure 2). With this intuition, we generate pseudo-labels by running a k-means clustering over the randomly selected columns to improve the diversity and the possibility of sampling highly correlated columns (with the original label).

Formally, to generate a single task $\mathcal{T}_{\texttt{STUNT}}$, we sample the masking ratio $p$ from the uniform distribution $U(r_1, r_2)$, where $r_1$ and $r_2$ are hyperparameters with $0 < r_1 < r_2 < 1$, and generate a random binary mask $\mathbf{m} := [m_1, \ldots, m_d]^\top \in \{0,1\}^d$ where $\sum_i m_i = \lfloor dp \rfloor$ and $\lfloor \cdot \rfloor$ is the floor function, i.e., the greatest integer not exceeding the input. Then, for a given unlabeled data $\mathcal{D}_u$, we select columns with the mask index with the value of one, i.e., $\texttt{sq}(\mathbf{x} \odot \mathbf{m}) \in \mathbb{R}^{\lfloor dp \rfloor}$ where $\odot$ is the element-wise product and $\texttt{sq}(\mathbf{x} \odot \mathbf{m})$ is a squeezing operation that removes the elements with the mask value of zero. Based on the selected columns, we run a k-means clustering to generate the task label $\tilde{\mathbf{y}}_{u,i}$:

$$\min_{\mathbf{C} \in \mathbb{R}^{\lfloor dp \rfloor \times k}} \frac{1}{N} \sum_{i=1}^N \min_{\tilde{\mathbf{y}}_{u,i} \in \{0,1\}^k} \|\texttt{sq}(\mathbf{x}_{u,i} \odot \mathbf{m}) - \mathbf{C}\tilde{\mathbf{y}}_{u,i}\|_2^2 \quad \text{such that} \quad \tilde{\mathbf{y}}_{u,i}^\top \mathbf{1}_k = 1, \quad (1)$$

where $k$ is the number of centroids, $\mathbf{1}_k \in \mathbb{R}^k$ is a vector of ones, and $\mathbf{C}$ is the centroid matrix.

Since the task label $\tilde{\mathbf{y}}_u$ is generated from the data itself, the classifier can easily infer the label from the given clean data $\mathbf{x}_u$. To prevent such an issue, we suggest perturbing the selected column features by $\tilde{\mathbf{x}}_u := \mathbf{m} \odot \hat{\mathbf{x}}_u + (1 - \mathbf{m}) \odot \mathbf{x}_u$ where each column feature element of $\hat{\mathbf{x}}_u$ is sampled from the empirical marginal distribution of each column feature. Finally, the generated task from STUNT is defined as follows: $\mathcal{T}_{\texttt{STUNT}} := \{\tilde{\mathbf{x}}_{u,i}, \tilde{\mathbf{y}}_{u,i}\}_{i=1}^{N_u}$.

**Meta-learning with STUNT.** Based on the generated task, we suggest to meta-learn the network by utilizing Prototypical Network (ProtoNet; Snell et al. (2017)): performs a non-parametric classifier on top of the network's embedding space. Specifically, ProtoNet learns this embedding space in which classification can be performed by computing distances to prototype vectors of each class, i.e., the average embedding vector of the class samples. The reason why we consider ProtoNet as a meta-learning framework is three-fold. First, the number of classes can be different for training and inference on ProtoNet, allows us to search the effective centroid number $k$ rather than fixing it to the class size $C$. Second, the method is model- and data-agnostic, where it can be used for the tabular domain without any modification. Finally, despite the simplicity, ProtoNet is known to outperform advanced meta-learning schemes under various datasets (Ye et al., 2020; Tack et al., 2022).

For a given task $\mathcal{T}_{\text{STUNT}}$, we sample two disjoint sets from $\mathcal{T}_{\text{STUNT}}$, i.e., $\mathcal{S}$ and $\mathcal{Q}$, which are used for constructing the classifier, and training the constructed classifier, respectively. Concretely, we construct the ProtoNet classifier $f_\theta$ over the parameterized embedding $z_\theta : \mathcal{X} \to \mathbb{R}^D$ by using the prototype vectors of each pseudo-class $\mathbf{p}_{\tilde{c}} := \frac{1}{|\mathcal{S}_{\tilde{c}}|} \sum_{(\tilde{\mathbf{x}}_u, \tilde{\mathbf{y}}_u) \in \mathcal{S}_{\tilde{c}}} z_\theta(\tilde{\mathbf{x}}_u)$ where $\mathcal{S}_{\tilde{c}}$ contains samples with pseudo-class $\tilde{c}$ in $\mathcal{S}$:

$$f_\theta(y = \tilde{c} | \mathbf{x}; \mathcal{S}) = \frac{\exp(-\|z_\theta(\mathbf{x}) - \mathbf{p}_{\tilde{c}}\|_2)}{\sum_{\tilde{c}'} \exp(-\|z_\theta(\mathbf{x}) - \mathbf{p}_{\tilde{c}'}\|_2)}. \tag{2}$$

We then compute the cross-entropy loss $\mathcal{L}_{\text{CE}}$ on the conducted classifier $f_\theta$ with set $\mathcal{Q}$, i.e., $\mathcal{L}_{\text{meta}}(\theta, \mathcal{Q}) := \sum_{(\tilde{\mathbf{x}}_u, \tilde{\mathbf{y}}_u) \in \mathcal{Q}} \mathcal{L}_{\text{CE}}(f_\theta(\tilde{\mathbf{x}}_u; \mathcal{S}), \tilde{\mathbf{y}}_u)$, where we train the network to minimize the meta-learning loss $\mathcal{L}_{\text{meta}}$ over the diverse set of tasks $\{\mathcal{T}_{\text{STUNT},1}, \mathcal{T}_{\text{STUNT},2}, \dots\}$.

**Adapting with labeled samples.** After meta-learning the parameter $\theta$ with self-generated tasks, we use the labeled set $\mathcal{D}_l$ to construct the classifier for the few-shot classification by using ProtoNet, i.e., $f_\theta(\cdot; \mathcal{D}_l)$ where the each prototype vector $\mathbf{p}_c$ is computed with samples of the label $c$ in $\mathcal{D}_l$.

### 3.3 PSEUDO-VALIDATION WITH STUNT

We find that the difficulty of the proposed unsupervised learning is the absence of a validation set for selecting the hyperparameters and early stopping the training. To tackle this issue, we introduce an unsupervised validation scheme where we generate a pseudo-validation set by running STUNT on the unlabeled set. Here, rather than sampling the columns for generating the cluster, we use all column features to remove the randomness throughout the validation and further use the clean tabular input contrary to the perturbed sample as in the original STUNT.

Formally, we sample a certain portion of the unlabeled set $\mathcal{D}_u^{\text{val}} \subset \mathcal{D}_u$, then generate the task label $\mathbf{y}_u^{\text{val}}$ by running a k-means clustering over clean samples $\mathbf{x}_u^{\text{val}} \in \mathcal{D}_u$ where $k = C$, i.e., Eq. (1) with $\mathbf{m} = \mathbf{1}_d$. Then, for a given validation task $\mathcal{T}_{\text{STUNT}}^{\text{val}} = \{\mathbf{x}_{u,i}^{\text{val}}, \mathbf{y}_{u,i}^{\text{val}}\}_i$, we sample two disjoint sets, $\mathcal{S}^{\text{val}}$ and $\mathcal{Q}^{\text{val}}$, to evaluate the pseudo-validation performance of the ProtoNet classifier $f_\theta(\cdot; \mathcal{S}^{\text{val}})$ by using Eq. (2), i.e., predicting pseudo-class of $\mathcal{Q}^{\text{val}}$ using prototype vectors made from $\mathcal{S}^{\text{val}}$.

## 4 EXPERIMENTS

In this section, we validate the effectiveness of our method on few-shot tabular learning scenarios under various tabular datasets from the OpenML-CC18 benchmark (Bischl et al., 2021). Our results exhibit that STUNT consistently and significantly outperforms other methods, including unsupervised, semi- and self-supervised methods (Section 4.1). We further demonstrate that our method is even effective for few-shot multi-task learning (Section 4.2). Finally, we perform an ablation study to verify the effect of the proposed pseudo-validation scheme of our approach (Section 4.3).

**Common setup.** For all the datasets, 80% of the data is used for training (unlabeled except for few-shot labeled samples) and 20% for testing, except for the income dataset, since split training and test data are provided. For STUNT, we use 20% of training data for pseudo-validation. We one-hot encode categorical features following the preprocessing of SubTab (Ucar et al., 2021) then apply normalization by subtracting the mean and dividing by the standard deviation for the income dataset and min-max scaling for other datasets, respectively. All baselines and STUNT are trained for 10K steps, while we follow the original training setting for CACTUs (Hsu et al., 2018). For all methods, we train a 2-layer multi-layer perceptron (MLP) with a hidden dimension of 1024. We provide additional information in the Appendix A.

Table 1: Few-shot test accuracy (%) on 8 datasets from the OpenML-CC18 benchmark (Bischl et al., 2021). We report the mean test accuracy over 100 different seeds. Checkmark ✓ indicates the use of 100 additional labeled samples for validation (Val.), including hyperparameter searching and early stopping. The bold denotes the highest mean score.

| Type | Method | Val. | income | cmc | karhunen | optdigit | diabetes | semeion | pixel | dna | Avg. |
|------|--------|------|--------|-----|----------|----------|----------|---------|-------|-----|------|
| | | | | | | # shot = 1 | | | | | |
| Sup. | CatBoost | ✓ | 57.00 | 34.60 | 55.67 | 61.32 | 60.02 | 43.21 | 59.16 | 41.35 | 52.06 |
| | MLP | ✓ | 60.52 | 35.06 | 48.67 | 61.02 | 57.25 | 40.88 | 55.62 | 44.39 | 50.43 |
| | LR | ✓ | 59.64 | 35.08 | 55.05 | 65.19 | 57.61 | 42.90 | 59.71 | 44.28 | 52.43 |
| | kNN | - | 61.22 | 34.99 | 54.42 | 65.58 | 58.56 | 44.35 | 61.48 | 42.67 | 52.82 |
| Semi-sup. | Mean Teacher | ✓ | 60.63 | 35.58 | 54.57 | 66.10 | 58.05 | 43.56 | 61.02 | 46.58 | 53.26 |
| | ICT | ✓ | 61.83 | 36.53 | 58.37 | 69.12 | 58.08 | 43.48 | 60.88 | 46.55 | 54.36 |
| | Pseudo-Label | ✓ | 60.52 | 34.97 | 49.44 | 61.50 | 57.03 | 41.42 | 56.12 | 44.26 | 50.66 |
| | MPL | ✓ | 60.85 | 35.13 | 47.66 | 61.52 | 57.39 | 41.82 | 56.01 | 44.22 | 50.58 |
| | VIME-Semi | ✓ | 56.40 | 32.97 | 57.40 | 66.85 | 58.16 | 40.43 | 52.86 | 39.18 | 50.53 |
| Self-sup. | SubTab + Fine-tune | ✓ | 59.74 | 35.65 | 41.11 | 49.88 | 59.35 | 30.49 | 42.23 | 40.86 | 44.91 |
| | SubTab + LR | ✓ | 61.88 | 35.68 | 50.32 | 67.05 | 58.06 | 40.27 | 60.40 | 45.68 | 52.42 |
| | SubTab + kNN | - | 61.58 | 35.87 | 48.74 | 66.05 | 59.22 | 39.99 | 61.30 | 44.16 | 52.36 |
| | VIME + Fine-tune | ✓ | 60.50 | 34.98 | 47.50 | 61.31 | 57.23 | 41.09 | 53.79 | 44.30 | 50.09 |
| | VIME + LR | ✓ | 61.99 | 35.30 | 59.62 | 70.52 | 56.95 | 47.20 | 64.17 | 51.36 | 55.89 |
| | VIME + kNN | - | 62.16 | 35.55 | 58.56 | 69.31 | 58.35 | 46.99 | 64.62 | 50.29 | 55.78 |
| Unsup.-Meta. | UMTRA | - | 57.23 | 35.46 | 49.05 | 49.87 | 57.64 | 26.33 | 34.26 | 25.13 | 41.87 |
| | SES | - | 56.39 | 34.59 | 49.19 | 56.30 | 59.97 | 33.73 | 49.19 | 39.56 | 47.37 |
| | CACTUs | - | **64.02** | 36.10 | 65.59 | 71.98 | 58.92 | 48.96 | 67.61 | 65.93 | 59.89 |
| | **STUNT (Ours)** | - | 63.52 | **37.10** | **71.20** | **76.94** | **61.08** | **55.91** | **79.05** | **66.20** | **63.88** |
| | | | | | | # shot = 5 | | | | | |
| Sup. | CatBoost | ✓ | 64.51 | 39.75 | 82.38 | 84.05 | 65.75 | 68.69 | 84.49 | 63.46 | 69.14 |
| | MLP | ✓ | 66.25 | 37.40 | 77.56 | 83.30 | 64.32 | 66.25 | 81.97 | 59.73 | 67.10 |
| | LR | ✓ | 66.53 | 37.15 | 81.02 | 86.22 | 64.19 | 67.87 | 85.02 | 58.88 | 68.36 |
| | kNN | - | 70.49 | 38.56 | 79.98 | 84.89 | 67.32 | 68.33 | 84.02 | 61.45 | 69.38 |
| Semi-sup. | Mean Teacher | ✓ | 67.05 | 37.73 | 81.08 | 86.66 | 65.45 | 69.67 | 85.24 | 61.47 | 69.29 |
| | ICT | ✓ | 70.13 | 38.09 | 84.58 | 87.01 | 65.47 | 70.26 | 86.12 | 63.37 | 70.63 |
| | Pseudo-Label | ✓ | 66.26 | 37.49 | 78.60 | 83.71 | 64.46 | 67.49 | 82.94 | 60.06 | 67.63 |
| | MPL | ✓ | 67.61 | 37.47 | 77.85 | 83.70 | 64.51 | 67.08 | 82.39 | 59.65 | 67.53 |
| | VIME-Semi | ✓ | 65.13 | 37.32 | 80.53 | 87.13 | 65.39 | 64.80 | 82.83 | 52.08 | 66.90 |
| Self-sup. | SubTab + Fine-tune | ✓ | 66.01 | 37.60 | 67.80 | 75.40 | 66.69 | 56.46 | 75.34 | 55.62 | 62.62 |
| | SubTab + LR | ✓ | 70.12 | 37.67 | 73.25 | 86.07 | 64.92 | 61.34 | 82.14 | 58.90 | 66.80 |
| | SubTab + kNN | - | 71.91 | 39.51 | 69.56 | 83.60 | 68.79 | 59.87 | 80.13 | 61.57 | 66.87 |
| | VIME + Fine-tune | ✓ | 65.97 | 37.25 | 77.82 | 83.13 | 64.40 | 63.63 | 81.01 | 59.58 | 66.60 |
| | VIME + LR | ✓ | 67.80 | 37.51 | 82.87 | 87.42 | 64.29 | 71.53 | 86.79 | 69.62 | 70.98 |
| | VIME + kNN | - | 72.16 | 39.28 | 79.15 | 83.86 | 69.45 | 71.53 | 84.07 | 71.09 | 70.63 |
| Unsup.-Meta. | UMTRA | - | 65.78 | 38.05 | 67.28 | 73.29 | 64.41 | 35.90 | 51.32 | 25.08 | 52.64 |
| | SES | - | 68.27 | 39.04 | 74.80 | 78.46 | 66.61 | 52.74 | 74.80 | 52.25 | 63.37 |
| | CACTUs | - | 72.03 | 38.81 | 82.20 | 85.92 | 66.79 | 65.00 | 85.25 | **81.52** | 72.19 |
| | **STUNT (Ours)** | - | **72.69** | **40.40** | **85.45** | **88.42** | **69.88** | **73.02** | **89.08** | 79.18 | **74.77** |

## 4.1 FEW-SHOT CLASSIFICATION

**Dataset.** We select 8 datasets from the OpenML-CC18 benchmark (Bischl et al., 2021; Asuncion & Newman, 2007). The income (Kohavi et al., 1996) and cmc dataset consists of both categorical and numerical features. The mfeat-karhunen (karhunen), optdigits, diabetes, semeion, mfeat-pixel (pixel) contain only numerical features. The dna dataset consists of only categorical features. We demonstrate the performance of STUNT on all types, as described in Appendix G. For dataset selection, we consider the following attributes: (i) whether the dataset consists of both categorical and numerical features, (ii) consists only of numerical or categorical features, (iii) type of task (i.e., binary classification or multi-way classification task). We validate that STUNT is generally applicable to arbitrary tabular data by performing experiments across datasets with the above properties.

**Baselines.** To validate our method, we compare the performance with four types of baselines: (i) supervised, (ii) semi-supervised, (iii) self-supervised, and (iv) unsupervised meta-learning methods. First, we compare with supervised learning methods such as CatBoost (Prokhorenkova et al., 2018), 2-layer MLP, k-nearest neighbors (kNN), and logistic regression (LR). kNN denotes the nearest neighbor classifier according to the prototype of the input data. Second, we compare our method to semi-supervised learning methods such as Mean Teacher (MT; Tarvainen & Valpola (2017)), Interpolation Consistency Training (ICT; Verma et al. (2019)), Pseudo-Label (PL; Lee (2013)), Meta Pseudo-Label (MPL; Pham et al. (2021)). We also have considered PAWS (Assran et al., 2021),

Table 2: 10-shot test accuracy (%) on 8 datasets from the OpenML-CC18 benchmark (Bischl et al., 2021). We report the mean test accuracy over 20 different seeds for each dataset. The bold indicates results within 1% from the highest mean score.

| Method \ Dataset | income | cmc | karhunen | optdigit | diabetes | semeion | pixel | dna | Avg. |
|---|---|---|---|---|---|---|---|---|---|
| kNN | **74.27** | 41.07 | 85.63 | 87.44 | 71.32 | 74.64 | 87.52 | 71.15 | 74.13 |
| ICT | 71.56 | 38.00 | **88.25** | **90.84** | 67.63 | 74.67 | **89.13** | 69.55 | 73.70 |
| VIME + LR | 69.17 | 37.92 | 86.63 | 89.63 | 66.56 | **77.66** | 88.71 | 74.73 | 73.88 |
| CACTUs | **73.63** | **42.14** | 85.48 | 87.92 | 70.75 | 68.22 | 87.21 | **84.40** | 74.97 |
| **STUNT (Ours)** | **74.08** | **42.01** | 86.95 | **89.91** | **72.82** | 74.74 | **89.90** | 80.96 | **76.42** |

the state-of-the-art semi-supervised learning method in the image domain, but observe that it is not effective for tabular datasets: we conjecture that the performance highly deviates from the choice of augmentation where tabular augmentation is not very effective compared to image augmentations. Third, we consider the recent state-of-the-art self-supervised method for tabular data, SubTab (Ucar et al., 2021) and VIME (Yoon et al., 2020) are pre-trained, then performance is evaluated with a few-shot labeled samples using fine-tuning, logistic regression, and kNN. Finally, we include CAC-TUs (Hsu et al., 2018), UMTRA (Khodadadeh et al., 2019), and SES (Ye et al., 2022) (along with semi-normalized similarity) as an unsupervised meta-learning baseline. Even though it is not clear how to design the augmentation strategy when applying UMTRA and SES to tabular data, we use marginal distribution masking, which are simple augmentation strategies used in SubTab. We provide additional results in Appendix I. We exclude Meta-GMVAE (Lee et al., 2021a) since in our experiments, it lags behind the baseline with the lowest performance that we consider. This is because training a variational auto-encoder (VAE; Kingma & Welling (2014)) for tabular datasets is highly non-trivial (Xu et al., 2019).

**Few-shot classification.** For the few-shot classification, we evaluate the performance when one and five labeled samples are available per class, respectively. We find that some baselines, such as Cat-Boost and ICT, require a validation set as they are highly sensitive to hyperparameters.Therefore, we perform a hyperparameter search and early stopping with 100 additional labeled samples for all baseline except for kNNs and unsupervised meta-learning methods. We note that using more labeled samples for validation than training is indeed unrealistic. On the other hand, we use the proposed pseudo-validation scheme for hyperparameter searching and early stopping of STUNT. One surprising observation is that CatBoost even lags behind kNN despite careful hyperparameter search. This implies that gradient boosting decision tree algorithms may fail in few-shot learning, while they are one of the most competitive models in fully-supervised settings (Shwartz-Ziv & Armon, 2022). In addition, semi-supervised learning methods achieve relatively low scores, which means that in tabular domain, pseudo-label quality goes down when the number of labeled samples is extremely small. Also, unlike the results of the image domain, UMTRA and SES perform worse than CACTUs. We believe that the failures of them are mainly due to the absence of effective augmentation strategies for tabular data.

As shown in Table 1, STUNT significantly improves the few-shot tabular classification performance even without using a labeled validation set. For instance, STUNT outperforms CACTUs from 67.61%→79.05% in the 1-shot classification of the pixel dataset. In particular, STUNT achieves the highest score in 7 of 8 datasets in both 1-shot and 5-shot classification problems, performing about 4% and 2% better than CACTUs in 1-shot and 5-shot cases, respectively. This is because STUNT is a *tabular-specific* unsupervised learning method that generates myriad meta-training tasks than CACTUs because we randomly select a subset of the column features for every training iteration.

**Low-shot classification.** We also validate our method when more labels are available, i.e., 10-shot. For baselines, we choose kNN, ICT, VIME + LR, and CACTUs because they show the best performance among supervised, semi-supervised, self-supervised, and unsupervised meta-learning methods in 1-shot and 5-shot classifications, respectively. Since a sufficient number of labeled training samples are available, we use the 2-shot sample from the 10-shot training sample for validation if the baseline requires hyperparameter search and early stop. In contrast, STUNT still does not use the labeled validation set, i.e., we utilize the proposed pseudo-validation scheme. As shown in Table 2, STUNT achieves the best score on average accuracy even under the low-shot classification setup.

While STUNT outperforms the baselines in the few- and low-shot learning setups, we find ensembles of decision trees or other semi-supervised learning methods, e.g., CatBoost or ICT, achieve a better

Table 3: Few-shot multi-task test accuracy (%) on the emotions dataset (Vanschoren et al., 2014), consists of 6 binary classification tasks. We report the mean test accuracy over 100 different seeds for each task. The bold indicates the highest mean score.

| Method \ Task | amazed-surprised | happy-please | relaxing-calm | quiet-still | sad-lonely | angry-aggressive | Avg. |
|---|---|---|---|---|---|---|---|
| # shot = 1 | | | | | | | |
| kNN | 59.04 | 47.14 | 55.77 | 66.86 | 55.96 | 59.47 | 57.37 |
| SubTab + kNN | **63.32** | 48.88 | 56.46 | 62.56 | 54.34 | 57.99 | 57.26 |
| VIME + kNN | 60.07 | 49.51 | 55.62 | 64.74 | 53.95 | 60.29 | 57.36 |
| CACTUs | 61.58 | 50.67 | 55.63 | 63.18 | 55.10 | 59.39 | 57.59 |
| **STUNT (Ours)** | 62.71 | **51.63** | **59.28** | **69.34** | **56.38** | **63.43** | **60.46** |
| # shot = 5 | | | | | | | |
| kNN | 70.71 | 53.48 | 66.34 | 81.03 | 68.51 | 68.07 | 68.02 |
| SubTab + kNN | **74.41** | 52.23 | 64.90 | 72.70 | 62.32 | 63.30 | 64.98 |
| VIME + kNN | 70.71 | 53.10 | 66.24 | 79.54 | 66.34 | 67.76 | 67.28 |
| CACTUs | 71.41 | 53.64 | 65.18 | 77.57 | 64.15 | 66.57 | 66.42 |
| **STUNT (Ours)** | 72.38 | **55.09** | **67.39** | **83.10** | **68.61** | **70.10** | **69.45** |

performance when more labeled samples are provided, e.g., 50-shot. Although the few-shot learning scenario is of our primary interest and our current method is specialized for the purpose, we think further improving our method under many-shot regimes would be an interesting future direction.

## 4.2 MULTI-TASK LEARNING

In this section, we introduce another application of STUNT, the few-shot multi-task learning. As STUNT learns to generalize across various self-generated tasks, it can instantly be adapted to multiple tasks at test-time without further training the network. Formally, we consider $\mathbf{x}_l$ and $\mathbf{x}_u$ are sampled from a same marginal distribution $p(\mathbf{x})$, where the label space $\mathcal{Y}$ differs.

**Dataset.** We use the emotions dataset from OpenML (Vanschoren et al., 2014), consisting of a variety of audio data with multiple binary labels and 72 numerical features. In particular, the emotion dataset aims to classify the multiple properties: amazed-surprised, happy-please, relaxing-calm, quiet-still, sad-lonely, or angry-aggressive. Because it is multi-labeled, data can have multiple attributes simultaneously, such as amazed-surprised and relaxing-calm audio.

**Baselines.** We compare STUNT with four baselines: kNN, SubTab + kNN, VIME + kNN, and CACTUs. All four baselines are chosen because they can adapt to multiple tasks with only one training procedure. On the other hand, methods such as ICT are excluded because they need a training procedure for each task. For example, in the case of the emotions dataset, six models are required.

**Multi-task learning.** As shown from Table 3, STUNT outperforms 5 out of 6 tasks in both 1-shot and 5-shot multi-task adaptation. In particular, STUNT outperforms the best baseline from 57.59%→60.46% in 1-shot multi-task, and 68.02%→69.45% in 5-shot. This is because STUNT generates a wide variety of meta-training tasks based on the fact that there are myriad ways to randomly select subsets of column features. In addition, it makes sense to treat column features as alternate labels, especially in tabular data, because each column feature has a different meaning. Considering the presence of critical real-world few-shot multi-task scenarios, such as patients with more than one disease, we conclude that STUNT is a promising way to mitigate these problems.

## 4.3 EFFECTIVENESS OF PSEUDO-VALIDATION

In this section, we perform further analysis of the proposed pseudo-validation with STUNT. For the analysis, we use four datasets from the OpenML-CC18 benchmark: two datasets containing both categorical and numerical features (i.e., income, cmc) and two datasets with only numerical features (i.e., semeion, pixel). We validate the model by constructing a number of 1-shot meta-validation tasks, i.e., $|\mathcal{S}^{\mathtt{val}}| = C$, with an unlabeled validation set for all experiments in this section.

**Hyperparameter search.** To validate that the proposed pseudo-validation scheme is useful for hyperparameter search, we show the correlation between the pseudo-validation accuracy and the test accuracy (achieved from the early stop point by the highest pseudo-validation accuracy). As shown in Figure 3, pseudo-validation accuracy and test accuracy have a positive correlation, which means

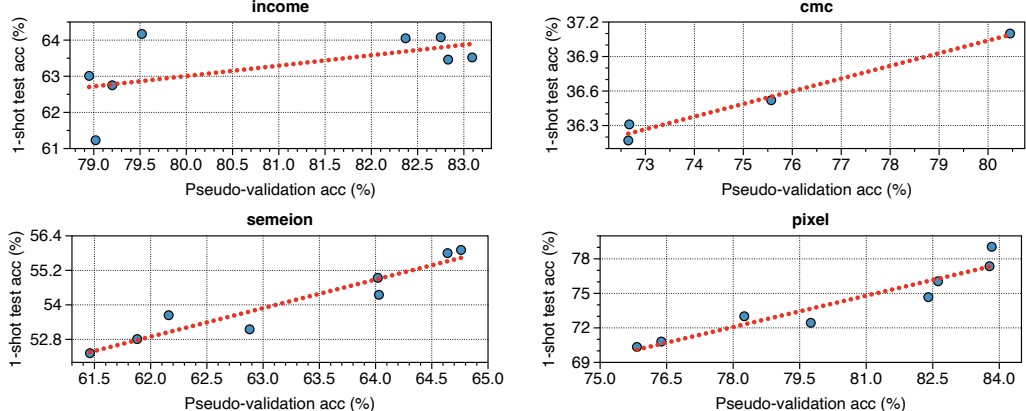

Figure 3: Correlation between the pseudo-validation accuracy (%) and the 1-shot test accuracy (%). The 1-shot test accuracy is achieved from the early stopped point by the highest pseudo-validation accuracy. Blue dots represent models trained with different hyperparameters. Red lines are the result of linear regression result of pseudo-validation accuracy and 1-shot test accuracy.

Table 4: Early stopping performance with the pseudo-validation set. We report 1-shot and 5-shot test accuracy (%) of fully trained (Last) and early stopped models (Early). We report the mean test accuracy of 100 different seeds. The bold indicates the highest mean score.

| Problem | income | | cmc | | semeion | | pixel | |
|---|---|---|---|---|---|---|---|---|
| | Last | Early | Last | Early | Last | Early | Last | Early |
| 1-shot | 61.58 | **63.52** | 36.94 | **37.10** | 51.94 | **55.91** | 74.92 | **79.05** |
| 5-shot | 70.84 | **72.69** | **40.43** | 40.40 | 71.55 | **73.02** | 87.60 | **89.08** |

that the higher the best pseudo-validation accuracy, the higher the test accuracy. Therefore, we use the pseudo-validation technique to search the hyperparameters of STUNT. Specifically, for the income, semeion, and pixel datasets, we find hyperparameters in eight combinations of hyperparameters. For the cmc dataset, we find hyperparameters in four combinations (indicated by blue dots in Figure 3). Although the best validation score may not guarantee the optimal hyperparameters, our method still gives reasonable hyperparameters. Additional information are reported in Appendix E.

**Early stopping.** As shown in Table 4, our pseudo-validation method is also useful for relaxing the overfitting issue. For example, on the pixel dataset, evaluating with the early stop model achieves 4.13% better accuracy than evaluating with the model after 10K training steps. Sometimes it is better to evaluate the model after a full training step, such as the cmc dataset, but our method still provides a reasonable early stopping rule when we see that the performance of the early stopped model by the highest pseudo-validation result only performs about 0.03% lower on the cmc dataset. In addition, the optimal required training steps are not known and often vary widely across datasets, especially in the tabular domain. For example, Levin et al. (2022) uses different fine-tuning epochs for different training setups (e.g., if 4 downstream training samples are available, use 30 fine-tuning epochs, and if 20 samples are available, use 60 fine-tuning epochs). However, since we use the pseudo-validation approach for early stopping, all we have to do is train the model for enough training steps (e.g., 10K training steps in our case) and use the model that achieves the best pseudo-validation score.

## 5 CONCLUSION

In this paper, we tackle the few-shot tabular learning problem, which is an under-explored but important research question. To this end, we propose STUNT, a simple yet effective framework that meta-learns over the self-generated tasks from unlabeled tables. Our key idea is to treat randomly selected columns as target labels to generate diverse few-shot tasks. The effectiveness of STUNT is validated by various few-shot classification tasks on different types of tabular datasets, and we also show that the representations extracted by STUNT apply well in multi-task scenarios. We hope that our work will guide new interesting directions in tabular learning field in the future.

## ETHICS STATEMENT

Since tabular data sometimes consists of privacy-sensitive features, e.g., social security number, one should always use the data carefully. However, STUNT is able to be well trained even with the encrypted data since the key idea is to use the tabular's unique property that each column has distinct meanings. Therefore, even though privacy issues may occur for tabular learning, STUNT has the potential to be generally used along with privacy-preserving techniques, such as homomorphic encryption (Cheon et al., 2017). Unsupervised meta-learning with STUNT on encrypted features is an interesting future direction.

## REPRODUCIBILITY STATEMENT

We provide code for reproduction in the supplementary material and describe the implementation details in Appendix B.

## ACKNOWLEDGMENTS AND DISCLOSURE OF FUNDING

We would like to thank Sangwoo Mo, Jaehyung Kim, Sihyun Yu, Subin Kim, and anonymous reviewers for their helpful feedbacks and discussions. This work was supported by Institute of Information & communications Technology Planning & Evaluation (IITP) grant funded by the Korea government (MSIT) (No.2019-0-00075, Artificial Intelligence Graduate School Program (KAIST); No.2022-0-00959, Few-shot Learning of Casual Inference in Vision and Language for Decision Making) and Samsung Electronics Co., Ltd (IO201211-08107-01).

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

## A    BASELINE DETAILS

In this section, we provide brief explanations of the considered baselines and the hyperparameter search space of the baselines. In common, we use Adam optimizer (Kingma & Ba, 2014) with learning rate $1e - 3$, and weight decay $1e - 4$. All baselines are trained with batch size 100 for all experiments.

**CatBoost.** CatBoost (Prokhorenkova et al., 2018) is one of gradient boosted decision tree algorithms. CatBoost consecutively builds decision trees in a way that reduces loss compared to previous trees. We search hyperparameters as shown in Table 5.

Table 5: Hyperparameter search space of Catboost (Prokhorenkova et al., 2018).

| Hyperparameter | Search space |
| --- | --- |
| iterations | {10, 100, 200, 500, 1000} |
| max depth | {4, 6, 8, 10} |
| learning rate | {0.001, 0.01, 0.1, 0.03} |
| bagging temperature | {0.6, 0.8, 1.0} |
| l2 leaf reg | {1, 3, 5, 7} |
| leaf estimation iterations | {1, 2, 4, 8} |

**Mean Teacher (MT).** MT (Tarvainen & Valpola, 2017) is semi-supervised learning method which uses the consistency loss between the teacher output and student output. The teacher model weights are updated as an exponential moving average of the student weights. We use the decay rate as 0.999 for exponential moving average. We search for weight of consistency loss in $\{0.1, 1, 10, 20, 50, 100\}$.

**Interpolation Consistency Training (ICT).** ICT (Verma et al., 2019) is a semi-supervised learning method uses MT framework while the student parameters are updated to encourage the consistency between the output of mixed samples and the mixed output of the samples. We use the decay rate as 0.999, and search for the weight of consistency loss in $\{0.1, 1, 10, 20, 50, 100\}$. We find the $\beta$ for Beta distribution in $\{0.1, 0.2, 0.5, 1\}$.

**Meta Pseudo-Label (MPL).** MPL (Pham et al., 2021) is a semi-supervised learning method which utilizes the performance of the student on the labeled dataset to inform the teacher to generate better pseudo-labels. In particular, the student model learns from pseudo-labeled data given from the teacher model. We use the decay rate as 0.999, and search for the weight of the unsupervised loss in $\{0.1, 1, 10, 20, 50, 100\}$.

**VIME.** VIME (Yoon et al., 2020) is a self-supervised learning method which extracts useful representation by corrupting random features and then predicting the corrupted location. For VIME pre-training, we follow the best hyperparameters suggested from original paper. Using VIME representations, we perform k-nearest neighbor classify, logistic regression, and fine-tuning. Early stop is done for logistic regression and fine-tuning.

**SubTab.** SubTab (Ucar et al., 2021) is a self-supervised learning method using effective three pretext task losses (i.e., reconstruction loss, contrastive loss, and distance loss). For SubTab pre-trianing, we follow the best hyperparameters suggested from original paper. Using SubTab representations, we perform k-nearest neighbor classify, logistic regression, and fine-tuning. Early stop is done for logistic regression and fine-tuning.

**CACTUs.** CACTUs (Hsu et al., 2018) is an unsupervised meta-learning method which runs a clustering algorithm on a representation trained with self-supervised learning in order to self-generate the tasks. We follow the hyperparameters suggested in the original paper.

For rest of the baselines, i.e., 2-layer multi-layer perceptron, logistic regression, pseudo-label (Lee, 2013), we use labeled validation set (i.e., additional 100 samples for 1-shot, 5-shot learning, and 2-shot samples for 10-shot learning) only for early stopping.

# B EXPERIMENTAL DETAILS

In this section, we provide hyperparameters of STUNT in Table 6 for each dataset found through the proposed pseudo-validation scheme. Shot indicates the number of sample per pseudo-class in $\mathcal{S}$, and query indicates the number of sample per pseudo-class in $\mathcal{Q}$. Way indicates the number of centroids in Eq. (1).

Table 6: Hyperparameters of STUNT

|         | income | cmc | karhunen | optdigit | diabetes | semeion | pixel | dna | emotions |
|---------|--------|-----|----------|----------|----------|---------|-------|-----|----------|
| # shot  | 1      | 1   | 1        | 1        | 1        | 1       | 1     | 1   | 1        |
| # query | 15     | 5   | 15       | 15       | 15       | 15      | 15    | 15  | 5        |
| # way   | 10     | 3   | 20       | 20       | 5        | 20      | 10    | 10  | 16       |

Except for shot, query, and way, we use full batch when self-generating STUNT tasks, and then use meta-training task batch size 4, $r_1 = 0.2$, $r_2 = 0.5$ and Adam optimizer with learning rate $1e - 3$.

# C DATASET DETAILS

In this section, we provide brief explanations of the considered datasets from the OpenML-CC18 benchmark (Vanschoren et al., 2014; Bischl et al., 2021).

**Income.** The task of the income (Kohavi et al., 1996; Bischl et al., 2021) dataset is to classify whether a person makes less than 50K a year or more than 50K a year.

**Cmc.** Cmc (Asuncion & Newman, 2007; Bischl et al., 2021) is an abbreviation for Contraceptive Method Choice. Literally, the target task is to predict the contraceptive method choice (i.e., no use, long-term or short-term).

**Mfeat-karhunen (karhunen), mfeat-pixel (pixel).** The karhunen and pixel (Asuncion & Newman, 2007; Bischl et al., 2021) datasets describe features of handwritten numbers. In particular, the karhunen dataset aims to find the correlation between 64 features obtained through the Karhunen-Loeve Transform and the 10 handwritten numbers drawn from the Dutch utility maps. On the other hand, the pixel dataset consists of 240 features by averaging $2 \times 3$ windows.

**Optdigit.** The optdigit (Asuncion & Newman, 2007; Bischl et al., 2021) is the dataset that describes the optical recognition of handwritten digits.

**Diabetes.** Literally, the diabetes (Asuncion & Newman, 2007; Bischl et al., 2021) dataset aims to predict whether the patient is tested positive for diabetes or not. In particular, the dataset features are composed of 8 numerical features, including diastolic blood pressure and body mass index.

**Semeion.** Semeion (Asuncion & Newman, 2007; Bischl et al., 2021) dataset is drawn by scanning and documenting handwritten digits from around 80 people.

**Dna.** The task of the dna (Bischl et al., 2021) dataset is to classify the boundaries between exons and introns with 180 indicator binary variables.

# D  ALGORITHM

---

**Algorithm 1** STUNT: Self-generated Tasks from UNlabeled Tables

---

**Require:** Unlabeled dataset $\mathcal{D}_u = \{\mathbf{x}_{u,i}\}_{i=1}^{N_u}$, Labeled dataset $\mathcal{D}_l = \{\mathbf{x}_{l,i}, \mathbf{y}_{l,i}\}_{i=1}^{N_l}$,
   task batch size $M_t$, learning rate $\beta$, mask ratio hyperparameters $r_1, r_2$.

---

1: Initialize $\theta$ using the standard initialization scheme.
2: // Step 1: Unsupervised meta-learning with STUNT
3: **while** not done **do**
4:  **for** $j = 1$ to $M_t$ **do**
5:   Sample mask ratio $p \sim U(r_1, r_2)$.
6:   $\mathbf{m} = [m_1, \ldots, m_d]^\top \in \{0, 1\}^d$ s.t. $\sum_i m_i = \lfloor dp \rfloor$.
7:   Run a k-means clustering: Eq. (1) with $\mathbf{x}_u \in \mathcal{D}_u$ and $\mathbf{m}$ to generate the task label $\tilde{\mathbf{y}}_u$.
8:   $\mathcal{T}_{\texttt{STUNT},j} = \{\tilde{\mathbf{x}}_{u,i}, \tilde{\mathbf{y}}_{u,i}\}_{i=1}^{N_u}$ where $\tilde{\mathbf{x}}_{u,i} := \mathbf{m} \odot \hat{\mathbf{x}}_{u,i} + (1 - \mathbf{m}) \odot \mathbf{x}_{u,i}$.
9:   Sample two disjoint sets $\mathcal{S}_j$ and $\mathcal{Q}_j$ from a given task $\mathcal{T}_{\texttt{STUNT},j}$.
10:   $\mathcal{L}_{\texttt{meta}}(\theta, \mathcal{Q}_j) = \sum_{(\tilde{\mathbf{x}}_u, \tilde{\mathbf{y}}_u) \in \mathcal{Q}_j} \mathcal{L}_{\texttt{CE}}\big(f_\theta(\tilde{\mathbf{x}}_u; \mathcal{S}_j), \tilde{\mathbf{y}}_u\big)$.
11:  **end for**
12:  $\theta \leftarrow \theta - \frac{\beta}{M_t} \cdot \nabla_\theta \sum_{j=1}^{M_t} \mathcal{L}_{\texttt{meta}}(\theta, \mathcal{Q}_j)$.
13: **end while**
14: // Step 2: Adapt classifier with the labeled dataset
15: Conduct a ProtoNet classifier $f_\theta(\cdot; \mathcal{D}_l)$ using $\mathcal{D}_l$.

---

# E  HYPERPARAMETER DETAILS OF ABLATION STUDY

As shown in Table 3, this section provides the search space of hyperparameters of ablation study in Section 4.3. Shot indicates the number of sample per pseudo-class in $\mathcal{S}$, and query indicates the number of sample per pseudo-class in $\mathcal{Q}$. Way indicates the number of centroids in Eq. (1).

Table 7: Hyperparameter search space of datasets used in Section 4.3

| Dataset | Hyperparameter | Search space |
|---------|----------------|--------------|
| income | (shot, query) | $\{(1, 5), (1, 15), (5, 10), (5, 20)\}$ |
|  | way | $\{5, 10\}$ |
| cmc | (shot, query) | $\{(1, 5), (1, 15), (5, 10), (5, 20)\}$ |
|  | way | $\{3\}$ |
| semeion | (shot, query) | $\{(1, 5), (1, 15), (5, 10), (5, 20)\}$ |
|  | way | $\{10, 20\}$ |
| pixel | (shot, query) | $\{(1, 5), (1, 15), (5, 10), (5, 20)\}$ |
|  | way | $\{10, 20\}$ |

## F  FEW-SHOT TABULAR REGRESSION RESULTS

Table 8: Few-shot regression tasks on 5 datasets from the OpenML (Vanschoren et al., 2014). We report the mean squared error over 100 different seeds for each dataset. The bold indicates the lowest mean error.

| Input | news | abalone | cholesterol | sarcos | boston |
|---|---|---|---|---|---|
| # shot = 5 | | | | | |
| Raw | 2.74E-04 | 1.75E-02 | 1.37E-02 | **1.05E-02** | 3.65E-02 |
| VIME | 2.69E-04 | 1.70E-02 | 1.37E-02 | 1.06E-02 | **3.53E-02** |
| CACTUs | 2.75E-04 | 1.72E-02 | 1.46E-02 | 1.06E-02 | 3.76E-02 |
| **STUNT (Ours)** | **2.68E-04** | **1.66E-02** | **1.35E-02** | 1.06E-02 | 3.70E-02 |
| # shot = 10 | | | | | |
| Raw | **2.53E-04** | 1.49E-02 | 1.13E-02 | 9.21E-03 | 2.88E-02 |
| VIME | **2.53E-04** | 1.49E-02 | 1.13E-02 | 9.24E-03 | **2.78E-02** |
| CACTUs | 2.54E-04 | 1.51E-02 | 1.21E-02 | **9.16E-03** | 2.94E-02 |
| **STUNT (Ours)** | **2.53E-04** | **1.46E-02** | **1.12E-02** | **9.16E-03** | 2.90E-02 |

We evaluate the capability of STUNT in few-shot regression tasks by replacing the ProtoNet classifier with a kNN regressor at the adaptation stage (i.e., after unsupervised meta-learning with STUNT).

We consider 5 tabular regression datasets in OpenML (Vanschoren et al., 2014), where we preprocess the input and target features with min-max scaling. For comparison, we evaluate the performance of the kNN regressor on VIME (Yoon et al., 2020) and CACTUs (Hsu et al., 2018) representations. Also, we report the performance of naive kNN regressor on the raw input. We use $k = 5$ and $k = 10$ for 5-shot and 10-shot experiments, respectively, where $k$ is the number of nearest neighbors.

In the Table 8, we report the average of mean-squared-errors (MSEs) over 100 different seeds of each method and dataset. The results indicate that STUNT is a competitive approach in a few-shot tabular regression task. However, the performance gap is often vacuous or marginal compared to the few-shot classification tasks. We believe that this is because STUNT meta-train networks with classification tasks, thus, can be more easily adapted to classification test-tasks. Therefore, extending STUNT by self-generating target-regression tasks with distinct column features could be effective in few-shot regression tasks, which we leave for future works.

## G  DATASET DESCRIPTION

Table 9: Dataset description. We select 8 tabular datasets from the OpenML-CC18 benchmark (Bischl et al., 2021) for extensive evaluation. The selected dataset consists of (i) both numerical and categorical features, (ii) only numerical features, and (iii) only categorical features.

| Property \ Dataset | income | cmc | karhunen | optdigits | diabetes | semeion | pixel | dna |
|---|---|---|---|---|---|---|---|---|
| # Columns | 14 | 9 | 64 | 64 | 8 | 256 | 240 | 180 |
| # Numerical | 6 | 2 | 64 | 64 | 8 | 256 | 240 | 0 |
| # Categorical | 8 | 7 | 0 | 0 | 0 | 0 | 0 | 180 |
| # Classes | 2 | 3 | 10 | 10 | 2 | 10 | 10 | 3 |

# H EFFECTIVENESS OF THE MARGINAL DISTRIBUTION MASKING

Table 10: Few-shot test accuracy (%) on 4 datasets from the OpenML-CC18 benchmark (Bischl et al., 2021) according to the masking type. We report the mean test accuracy over 100 different seeds. The bold denotes the highest mean score.

| Masking type | income | cmc | semeion | pixel | Avg. |
|---|---|---|---|---|---|
| # shot = 1 | | | | | |
| No-masking | 59.18 | 36.23 | 54.04 | 75.99 | 56.36 |
| Zero-masking | 61.88 | 35.47 | 54.44 | 77.49 | 57.32 |
| Gaussian noise | 60.34 | 36.49 | 55.45 | 78.39 | 57.67 |
| **Marginal distribution** | **63.52** | **37.10** | **55.91** | **79.05** | **58.90** |
| # shot = 5 | | | | | |
| No-masking | 70.94 | 38.76 | 72.12 | 86.38 | 67.05 |
| Zero-masking | 71.25 | 40.36 | 71.46 | 87.62 | 67.67 |
| Gaussian noise | 69.91 | 40.04 | **73.64** | 87.93 | 67.88 |
| **Marginal distribution** | **72.69** | **40.40** | 73.02 | **89.08** | **68.80** |

We compare marginal distribution masking with widely used masking strategies in the tabular domain; the zero-masking (i.e., replace the masked column feature with a zero value) and Gaussian noise (i.e., add gaussian noise to the masked column feature used in SubTab (Ucar et al., 2021)). For comparisons, we also use 4 datasets from the OpenML-CC18 benchmark (Bischl et al., 2021) that are used for experiments in Section 4.3. As shown in the Table 10, all masking strategies show meaningful improvement over the no-masking case, where our marginal distribution masking shows the best result.

Note that marginal distribution masking is a popular masking (and augmentation) scheme in many tabular models, such as SubTab (Ucar et al., 2021), VIME (Yoon et al., 2020), and SCARF (Bahri et al., 2022). On the other hand, zero-masking and Gaussian noise may have higher chances of generating unrealistic data points. For example, zero-masking makes the data too sparse, and adding Gaussian noise to one-hot encoded categorical features is unrealistic.

# I  COMPARISON WITH AUGMENTATION-BASED UNSUPERVISED META-LEARNING SCHEMES

Table 11: Few-shot test accuracy (%) on 8 datasets from the OpenML-CC18 benchmark (Bischl et al., 2021). We report the mean test accuracy over 100 different seeds. The bold denotes the highest mean score.

| Method | income | cmc | karhunen | optdigit | diabetes | semeion | pixel | dna | Avg. |
|---|---|---|---|---|---|---|---|---|---|
| # shot = 1 | | | | | | | | | |
| UMTRA + Gaussian noise | 60.15 | 34.37 | 47.80 | 38.85 | 58.38 | 25.00 | 32.77 | 23.25 | 40.07 |
| UMTRA + Marginal distribution masking | 57.23 | 35.46 | 49.05 | 49.87 | 57.64 | 26.33 | 34.26 | 25.13 | 41.87 |
| SES + Gaussian noise | 58.85 | 34.98 | 38.95 | 57.63 | 59.45 | 36.38 | 40.99 | 35.80 | 45.38 |
| SES + Marginal distribution masking | 56.39 | 34.59 | 49.19 | 56.30 | 59.97 | 33.73 | 49.19 | 39.56 | 47.37 |
| CACTUs | **64.02** | 36.10 | 65.59 | 71.98 | 58.92 | 48.96 | 67.61 | 65.93 | 59.89 |
| **STUNT (Ours)** | 63.52 | **37.10** | **71.20** | **76.94** | **61.08** | **55.91** | **79.05** | **66.20** | **63.88** |
| # shot = 5 | | | | | | | | | |
| UMTRA + Gaussian noise | 64.90 | 36.59 | 68.06 | 58.91 | 64.27 | 32.48 | 50.14 | 23.20 | 49.82 |
| UMTRA + Marginal distribution masking | 65.78 | 38.05 | 67.28 | 73.29 | 64.41 | 35.90 | 51.32 | 25.08 | 52.64 |
| SES + Gaussian noise | 64.28 | 38.70 | 60.50 | 77.55 | 67.32 | 56.70 | 57.96 | 40.39 | 57.93 |
| SES + Marginal distribution masking | 68.27 | 39.04 | 74.80 | 78.46 | 66.61 | 52.74 | 74.80 | 52.25 | 63.37 |
| CACTUs | 72.03 | 38.81 | 82.20 | 85.92 | 66.79 | 65.00 | 85.25 | **81.52** | 72.19 |
| **STUNT (Ours)** | **72.69** | **40.40** | **85.45** | **88.42** | **69.88** | **73.02** | **89.08** | 79.18 | **74.77** |

We evaluate UMTRA (Khodadadeh et al., 2019) and SES (Ye et al., 2022) (also utilizing SNS proposed by Ye et al. (2022)) on few-shot tabular learning tasks, where we use augmentation strategies used in SubTab (Ucar et al., 2021) (i.e., Gaussian noise and marginal distribution masking). Here, we tried our best to improve the performance of SES and UMTRA (e.g., tune variance of Gaussian noise). However, unlike the image domain, they performed worse than CACTUs (Hsu et al., 2018), as shown in Table 11. We believe that the failures of SES and UMTRA are mainly due to the absence of effective augmentation strategies for tabular data, and developing them will be an interesting future direction.

