# OpenReview forum: "STUNT: Few-shot Tabular Learning with Self-generated Tasks from Unlabeled Tables"
_ICLR.cc/2023/Conference — ICLR 2023 notable top 25%_

### Official Review · Reviewer_7dUE · 2022-10-24

**Confidence:** 3
**Correctness:** 3
**Technical Novelty And Significance:** 3
**Empirical Novelty And Significance:** 3
**Recommendation:** 6

**Clarity, Quality, Novelty And Reproducibility:**

Clarity: Good.

Quality: Good.

Novelty: Good.

Reproducibility: Code provided, hyperparameters clearly stated.

**Strength And Weaknesses:**

Strength:
  - This paper is well-written and easy to follow. One can understand the proposed method by simply looking at figure 1.
  - The proposed method is simple yet technically sound.
  - The empirical analysis shows the proposed method leads to a good performance boost over the baselines. The results are reliable since it's calculated over a lot of random seeds.

Weakness:
  - Since the method is proposed for `tabular learning` and  `regression` tasks are very common in tabular learning, I am curious to see how this method performs on tabular regression tasks.

**Summary Of The Paper:**

This paper introduces a new self-supervised training method for tabular learning, which leads to good performance improvements in the empirical analysis.

**Summary Of The Review:**

This paper presents a novel self-supervised learning method for few-shot tabular classification tasks. The method is simple and effective. The empirical results are convincing.

---

> ### Author Response · Authors · 2022-11-13
> **Response to Reviewer 7dUE**
>
> Dear reviewer 7dUE,
>
> We sincerely appreciate your efforts and insightful comments to improve the manuscript. We respond to each of your comments one-by-one in what follows. In the revised draft, we mark our revisions in “$\color{blue}{\text{blue}}$”.
>
> ----
>
> **[W1] I am curious to see how this method performs on tabular regression tasks.**
>
> Thank you for your constructive suggestion. To address your concern, we evaluate the capability of STUNT in few-shot regression tasks by replacing the ProtoNet classifier with a kNN regressor at the adaptation stage (i.e., after unsupervised meta-learning with STUNT).
>
> We consider 5 tabular regression datasets in OpenML, where we preprocess the input and target features with min-max scaling. For comparison, we evaluate the performance of the kNN regressor on VIME and CACTUs representations. Also, we report the performance of naive kNN regressor on the raw input. We use $k=5$ and $k=10$ for 5-shot and 10-shot experiments, respectively, where $k$ is the number of nearest neighbors. In the revised draft (see Appendix F), we included the experimental details and results on few-shot regression tasks.
>
> In the table below, we report the average of mean-squared-errors (MSEs) over 100 different seeds of each method and dataset. The results indicate that STUNT is a competitive approach in a few-shot tabular regression task. However, the performance gap is often vacuous or marginal compared to the few-shot classification tasks. We believe that this is because STUNT meta-train networks with classification tasks, thus, can be more easily adapted to classification test-tasks. Therefore, extending STUNT by self-generating target-regression tasks with distinct column features could be effective in few-shot regression tasks, which we leave for future works.
>
> \begin{array}{lccccc}
> \hline
> \text{Representation} & \text{news} & \text{abalone} & \text{cholesterol} & \text{sarcos} & \text{boston} \newline
> \hline
> & & \text{\\# shot = 5} & & & \newline
> \hline
> \text{Raw} & \text{2.74E-04} & \text{1.75E-02} & \text{1.37E-02} & \textbf{1.05E-02} & \text{3.65E-02} \newline
> \text{VIME} & \text{2.69E-04} & \text{1.70E-02} & \text{1.37E-02} & \text{1.06E-02} & \textbf{3.53E-02} \newline
> \text{CACTUs} & \text{2.75E-04} & \text{1.72E-02} & \text{1.46E-02} & \text{1.06E-02} & \text{3.76E-02} \newline
> \hline
> \textbf{STUNT (Ours)} & \textbf{2.68E-04} & \textbf{1.66E-02} & \textbf{1.35E-02} & \text{1.06E-02} & \text{3.70E-02} \newline
> \hline
> & & \text{\\# shot = 10} & & & \newline
> \hline
> \text{Raw} & \textbf{2.53E-04} & \text{1.49E-02} & \text{1.13E-02} & \text{9.21E-03} & \text{2.88E-02} \newline
> \text{VIME} & \textbf{2.53E-04} & \text{1.49E-02} & \text{1.13E-02} & \text{9.24E-03} & \textbf{2.78E-02} \newline
> \text{CACTUs} & \text{2.54E-04} & \text{1.51E-02} & \text{1.21E-02} & \textbf{9.16E-03} & \text{2.94E-02} \newline
> \hline
> \textbf{STUNT (Ours)} & \textbf{2.53E-04} & \textbf{1.46E-02} & \textbf{1.12E-02} & \textbf{9.16E-03} & \text{2.90E-02} \newline
> \hline
> \end{array}

---

> ### Author Response · Authors · 2022-12-06
> **A Gentle Reminder**
>
> Dear reviewer 7dUE,
>
> Thank you very much again for your time and efforts in reviewing our paper.
>
> We kindly remind that we have only a week or so in the discussion period.
>
> We just wonder whether there is any further concern and hope to have a chance to respond before the discussion phase ends.
>
> Many thanks, Authors

---

### Official Review · Reviewer_sZ8N · 2022-10-25

**Confidence:** 4
**Correctness:** 4
**Technical Novelty And Significance:** 3
**Empirical Novelty And Significance:** 3
**Recommendation:** 5

**Clarity, Quality, Novelty And Reproducibility:**

The main idea is clear. The usage of "few-shot" is a bit confusing since the method is introduced based on the semi-supervised learning setting. Although the authors apply STUNT on other few-shot applications, it should be made clear in the beginning of the paper.


**Strength And Weaknesses:**

This paper proposes a novel aspect of semi-supervised tabular data learning, which shows that unsupervised meta-learning is an effective tool for this task. The idea of task generation from unlabeled tables is novel and interesting. The experiments show the method could be generalized in various cases.

Here are some suggestions and questions for the paper:
1. There is another thread of unsupervised meta-learning such as [1] and [2]. The experiments in these papers show efficient unsupervised meta-learning works better than CACTUS and UMTRA. The authors need to discuss this method in the related work part and try to compare with them in experiments. [1] also shows the requirement of plenty of tasks in unsupervised meta-learning. How will the diversity of the generated tasks influence the convergence of the model?
2. The experiments show the unsupervised task works better than the contrastive strategy such as SubTab. Could the authors explain the reason why unsupervised meta-learning is better? Could we combine these strategies together?
3. The authors mentioned "We also exclude UMTRA (Khodadadeh
et al., 2019) from the baseline because it is not clear how to design the augmentation strategy for tabular data (UMTRA uses image augmentations)". Could the augmentation strategy in SubTab be applied with UMTRA and other unsupervised meta-learning methods?
4. In the step "adapting with labeled samples", does the model be fine-tuned on the labeled samples or only construct the prototypes directly?
5. How will the model performance changes w.r.t. different numbers of shots?

[1] Ye, Han, and Zhan. Revisiting Unsupervised Meta-Learning via the Characteristics of Few-Shot Tasks. TPAMI.
[2] Chen, Maji, and Learned-Miller, Shot in the dark: Few-shot learning with no base-class labels. CoRR.

**Summary Of The Paper:**

This paper proposes the Self-generated Tasks from UNlabeled Tables (STUNT) for few-shot tabular learning, where there are only a limited number of labeled training examples. STUNT applies unsupervised meta-learning to learn generalized knowledge, and an unsupervised validation strategy is also proposed. Experiments on various few-shot tabular data learning benchmarks validate the effectiveness of the method.


**Summary Of The Review:**

The paper is overall novel and interesting. Experiments show some convincing results. Some additional experiments and discussions are required to make the paper more solid. Please check the weakness part.

---

> ### Author Response · Authors · 2022-11-13
> **Response to Reviewer sZ8N (2/2)**
>
> **[W1-2] Influence of the diversity of the generated tasks (to the model convergence).**
>
> Typically, the higher the diversity of generated tasks, the slower (and more difficult) convergence of the model. However, learning from diverse tasks leads to better generalization after convergence.
>
> To address your concerns more, we conduct an ablation study by selecting a fixed set of columns (e.g., 30% of columns) when generating tasks to reduce the diversity. As a result, if the generated tasks were more diverse, the training loss decreased slowly, and the training accuracy also increased slowly (e.g., after training 2K steps, the training accuracy was 92.91% for low diversity, while 65.38% for high diversity). However, as shown in the table below, the test accuracy verifies that the generalization performance is better when trained with more various tasks. This result also highlights the importance of generating diverse meta-training tasks for few-shot tabular learning, which is indeed the main strength of STUNT compared to other unsupervised meta-learning schemes.
>
> \begin{array}{lccccc}
> \hline
> \text{Diversity} & \text{income} & \text{cmc} & \text{semeion} & \text{pixel} & \text{Avg.} \newline
> \hline
> & & \text{\\# shot = 1} & & & \newline
> \hline
> \text{Low} & 55.17 & 36.49 & 35.91 & 52.01 & 44.90 \newline
> \textbf{High (Ours)} & \textbf{63.52} & \textbf{37.10} & \textbf{55.91} & \textbf{79.05} & \textbf{58.90} \newline
> \hline
> & & \text{\\# shot = 5} & & & \newline
> \hline
> \text{Low} & 62.19 & 40.04 & 49.79 & 67.70 & 54.93 \newline
> \textbf{High (Ours)} & \textbf{72.69} & \textbf{40.40} & \textbf{73.02} & \textbf{89.08} & \textbf{68.80} \newline
> \hline
> \end{array}
>
> ----
>
> **[W2] Could the authors explain the reason why unsupervised meta-learning is better than the contrastive strategy such as SubTab? Could we combine these strategies together?**
>
> In our experiments, contrastive strategies do not bring meaningful performance gains for few-shot tabular learning (see Table 1 in our main draft). Hence, combining the method with poor performance does not necessarily help improve performance.
>
> The reason why contrastive strategies do not work in our tabular setups is that (i) the lack of effective augmentation and (ii) the gap between trained self-supervised tasks and the applied few-shot task is large due to the heterogeneous characteristics of tabular data. On the other hand, we utilize the power of meta-learning to reduce the gap via fast adaptation to unseen few-shot tasks.
>
> ----
>
> **[W4] Clarification of the “adapting with labeled samples”: does the model be fine-tuned on the labeled samples or only construct the prototypes directly?**
>
> We constructed the classifier with the prototypes of the given labeled samples (i.e., ProtoNet classifier) and did not finetune the classifier, because fine-tuning with few samples can easily overfit (without the labeled validation set).
>
> ----
>
> **[W5] How will the model performance change w.r.t. the different number of shots?**
>
> In our experiments, STUNT consistently achieves better test accuracy (than baselines) by adapting various numbers of few shots (see Table 1 and 2 in the main draft). Here, we also found that the model trained with 1-shot slightly (and consistently) showed a better result than that with other shots, irrespectively of the number of shots used in meta-testing (see the table below). This is even better for practitioners as one does not need to re-train STUNT again when the number of shots is changing. Note that it is actually a common observation in the unsupervised meta-learning domain, where a model trained with 1-shot can well generalize on new tasks [4, 5].
>
> \begin{array}{lccccc}
> \hline
> \text{\\# train shot} & \text{income} & \text{cmc} & \text{semeion} & \text{pixel} & \text{Avg.} \newline
> \hline
> & & \text{\\# test shot = 1} & & & \newline
> \hline
> \text{\\# train shot = 5} & 62.99 & 36.37 & 51.32 & 76.30 & 56.75\newline
> \textbf{\\# train shot = 1 (Ours)} & \textbf{63.52} & \textbf{37.10} & \textbf{55.91} & \textbf{79.05} & \textbf{58.90} \newline
> \hline
> & & \text{\\# test shot = 5} & & & \newline
> \hline
> \text{\\# train shot = 5} & 71.79 & 39.83 & 70.90 & 89.00 & 67.88 \newline
> \textbf{\\# train shot = 1 (Ours)} & \textbf{72.69} & \textbf{40.40} & \textbf{73.02} & \textbf{89.08} & \textbf{68.80} \newline
> \hline
> \end{array}
>
> [4] Hsu et al., “Unsupervised learning via meta-learning”, ICLR 2018
>
> [5] Khodadadeh et al., “Unsupervised meta-learning for few-shot image classification”, NeurIPS 2019
>
> ----
>
> **[C1] The usage of “few-shot” is a bit confusing**
>
> Thank you very much for the suggestion. In the revised manuscript, we clarified the major target of the paper is few-shot semi-supervised learning in both the abstract and introduction.

---

> ### Author Response · Authors · 2022-11-13
> **Response to Reviewer sZ8N (1/2)**
>
> Dear reviewer sZ8N,
>
> We sincerely appreciate your efforts and insightful comments to improve the manuscript. We respond to each of your comments one-by-one in what follows. In the revised draft, we mark our revisions in “$\color{blue}{\text{blue}}$”.
>
> ----
>
> **[W1-1, W3] Comparison (and discussions) with other unsupervised meta-learning schemes such as [1, 2]. Could the augmentation strategy in SubTab be applied with UMTRA and other unsupervised meta-learning methods?**
>
> Thank you for your constructive suggestion. In the revised draft (see Section 2, Section 4.1, Appendix I, and Table 1), we included the comparison (and discussions) with SES [1], UMTRA, and [2].
>
> To address your concerns, we evaluate UMTRA and SES (also utilizing SNS proposed by [1]) on few-shot tabular learning tasks, where we use augmentation strategies used in SubTab (i.e., Gaussian noise and marginal distribution masking). Here, we tried our best to improve the performance of SES and UMTRA (e.g., tune variance of Gaussian noise). However, unlike the image domain, they performed worse than our baseline, CACTUs, as shown in the table below. As we mentioned in the draft, we believe that the failures of SES and UMTRA are mainly due to the absence of effective augmentation strategies for tabular data, and developing them (e.g., finding appropriate augmentations automatically for each dataset [3]) will be an interesting future direction.
>
> \begin{array}{lccccccccc}
> \hline
> \text{Method} & \text{income} & \text{cmc} & \text{karhunen} & \text{optdigit} & \text{diabetes} & \text{semeion} & \text{pixel} & \text{dna} & \text{Avg.}\newline
> \hline
> & & & \text{\\# shot = 1} & & & & & & \newline
> \hline
> \text{UMTRA + Gaussian} & 60.15 & 34.37 & 47.80 & 38.85 & 58.38 & 25.00 & 32.77 & 23.25 & 40.07 \newline
> \text{UMTRA + Marginal distribution} & 57.23 & 35.46 & 49.05 & 49.87 & 57.64 & 26.33 & 34.26 & 25.13 & 41.87 \newline
> \text{SES + Gaussian} & 58.85 & 34.98 & 38.95 & 57.63 & 59.45 & 36.38 & 40.99 & 35.80 & 45.38 \newline
> \text{SES + Marginal distribution} & 56.39 & 34.59 & 49.19 & 56.30 & 59.97 & 33.73 & 49.19 & 39.56 & 47.37\newline
> \text{CACTUs} & \textbf{64.02} & 36.10 & 65.59 & 71.98 & 58.92 & 48.96 & 67.61 & 65.93 & 59.89\newline
> \hline
> \textbf{STUNT (Ours)} & 63.52 & \textbf{37.10} & \textbf{71.20} & \textbf{76.94} & \textbf{61.08} & \textbf{55.91} & \textbf{79.05} & \textbf{66.20} & \textbf{63.88} \newline
> \hline
> & & & \text{\\# shot = 5} & & & & & & \newline
> \hline
> \text{UMTRA + Gaussian} & 64.90 & 36.59 & 68.06 & 58.91 & 64.27 & 32.48 & 50.14 & 23.20 & 49.82 \newline
> \text{UMTRA + Marginal distribution} & 65.78 & 38.05 & 67.28 & 73.29 & 64.41 & 35.90 & 51.32 & 25.08 & 52.64 \newline
> \text{SES + Gaussian} & 64.28 & 38.70 & 60.50 & 77.55 & 67.32 & 56.70 & 57.96 & 40.39 & 57.93 \newline
> \text{SES + Marginal distribution} & 68.27 & 39.04 & 74.80 & 78.46 & 66.61 & 52.74 & 74.80 & 52.25 & 63.37\newline
> \text{CACTUs} & 72.03 & 38.81 & 82.20 & 85.92 & 66.79 & 65.00 & 85.25 & \textbf{81.52} & 72.19 \newline
> \hline
> \textbf{STUNT (Ours)} & \textbf{72.69} & \textbf{40.40} & \textbf{85.45} & \textbf{88.42} & \textbf{69.88} & \textbf{73.02} & \textbf{89.08} & 79.18 & \textbf{74.77}\newline
> \hline
> \end{array}
>
> [1] Ye et al., “Revisiting Unsupervised Meta-Learning via the Characteristics of Few-Shot Tasks, TPAMI 2022
>
> [2] Chen et al., “Shot in the Dark: Few-Shot Learning With No Base-Class Labels”, CVPRW 2021
>
> [3] Cubuk et al., “AutoAugment: Learning Augmentation Policies from Data”, CVPR 2019

---

> ### Author Response · Authors · 2022-12-06
> **A Gentle Reminder**
>
> Dear reviewer sZ8N,
>
> Thank you very much again for your time and efforts in reviewing our paper.
>
> We kindly remind that we have only a week or so in the discussion period.
>
> We just wonder whether there is any further concern and hope to have a chance to respond before the discussion phase ends.
>
> Many thanks, Authors

---

### Official Review · Reviewer_26Xs · 2022-10-27

**Confidence:** 3
**Correctness:** 4
**Technical Novelty And Significance:** 3
**Empirical Novelty And Significance:** 2
**Recommendation:** 8

**Clarity, Quality, Novelty And Reproducibility:**

Clarity is overall good. The method is mathematically described with sufficient precision. The novelty of the method is somewhat low as many techniques for generating meta-learning tasks from unsupervised data have been explored recently. This is counterbalanced to some extent by the application area of tabular data, which is relatively underexplored. Quality is good and the claims of the paper appear to be supported by experiments. As for reproducibility, code is included and experiments are described in sufficient detail.

Minor comment: Eq (2) appears to be missing negations inside the exps.

**Strength And Weaknesses:**

Strengths
- The proposed method is simple and intuitive.
- Experiments are fairly extensive and compare against a variety of baselines on several datasets and runs over multiple random seeds.
- Limitations of the method are discussed with respect to low-shot learning.

Weaknesses
- Novelty is somewhat low as the method could be viewed as a minor modification to CACTUs.
- There are a few design decisions that are not well justified: the use of all columns for clustering in the pseudo-validation, the effect of marginal distribution masking versus other types of masking, and how the column masking ratio should be chosen.

**Summary Of The Paper:**

This paper proposes an approach to few-shot learning with tabular data called STUNT, which creates synthetic meta-learning tasks from unlabeled tabular data by clustering randomly selected subsets of columns to form pseudo-labels. A pseudo-validation method is also proposed that can be used to perform early-stopping using the unlabeled data only. Experiments on a collection of tabular datasets representing a mix of numerical and categorical columns demonstrate that STUNT exhibits strong performance relative to a wide variety of baselines.

**Summary Of The Review:**

Post-rebuttal update: I would like to thank the authors for responding to my concerns. The paper has been improved quite a bit during the rebuttal phase. I appreciate the additional results on regarding the distribution masking, extra baselines as suggested by RsZ8N, and extensions to regression tasks as suggested by R7dUE. These changes have made the paper more complete, and I will update my rating accordingly.

---

This paper addresses an interesting problem, i.e. few-shot learning of tabular data, and proposes a simple yet effective solution to address it. The results appear to yield a significant improvement relative to baselines. The weakest point of the submission is novelty with respect to previous work on task generation from unsupervised data. However, the proposed method elegantly adapts previous work to the tabular setting and achieves good results.

---

> ### Author Response · Authors · 2022-11-13
> **Response to Reviewer 26Xs (2/2)**
>
> **[W2-3] Justification of the design choice: how to chose masking ratio.**
>
> The masking ratio of STUNT is not sensitive to its performance. In particular, we chose the masking ratio in the interval [0.2, 0.5] uniformly at random, which worked well consistently across all datasets tested. Remark that the effective choice of masking ratio varies by different datasets and tasks, for example, BERT [4] uses 0.15, and MAE [5] uses 0.75. One can tune this hyperparameter for each dataset to achieve an even better performance of our method, but we did not put such efforts in this paper.
>
> [4] Devlin et al., “BERT: Pre-training of Deep Bidirectional Transformers for Language Understanding”, NAACL 2019
>
> [5] He et al., “Masked Autoencoders Are Scalable Vision Learners”, CVPR 2022
>
> ----
>
> **[C1] Editorial comment**
>
> Thank you very much. We have fixed the typo in our revised manuscript: add the negation operator in Eq. (2). (Section 3.2)

---

> ### Author Response · Authors · 2022-11-13
> **Response to Reviewer 26Xs (1/2)**
>
> Dear reviewer 26Xs,
>
> We sincerely appreciate your efforts and insightful comments to improve the manuscript. We respond to each of your comments one-by-one in what follows. In the revised draft, we mark our revisions in “$\color{blue}{\text{blue}}$”.
>
> ----
>
> **[W1] Novelty is somewhat low: minor modification to CACTUs.**
>
> Our key novelty (different from CACTUs) is on how STUNT generates meta-training tasks, specialized to tabular data. To be specific, it self-generates a diverse set of meta-training tasks from the unlabeled tabular data by treating the table's column feature as a useful target. Despite its simplicity, such an approach is well-suited due to the heterogeneous nature of the tabular dataset; one can consider any column features as a task label. On the other hand, the way how CACTUs generates meta-training tasks is completely different from STUNT. In particular, to generate task labels, CACTUs always performs clustering on a (fixed) self-supervised representation space, while STUNT performs clustering on diverse/dynamic spaces formed by random subsets of column features. Hence, compared to CACTUs, STUNT makes great use of heterogeneous properties of tabular data by utilizing more diverse, yet realistic task labels. Furthermore, we also propose an unsupervised validation scheme by utilizing STUNT to the unlabeled set for hyperparameter selection and early stopping, which does not exist in CACTUs.
>
> ----
>
> **[W2-1] Justification of the design choice: use of all columns for clustering in the pseudo-validation.**
>
> We would like to clarify that we use the entire column features to remove the randomness during the pseudo-validation, which reduces its variance and thus computation cost. While randomly selecting the column features plays a crucial role in generating diverse meta-training tasks, it hampers meta-validation as it increases the possibility of choosing underperforming checkpoints due to randomness.
>
> ----
>
> **[W2-2] Justification of the design choice: other types of masking.**
>
> To address your concern, we compare marginal distribution masking with widely used masking strategies in the tabular domain; the zero-masking (i.e., replace the masked column feature with a zero value) and Gaussian noise (i.e., add gaussian noise to the masked column feature used in SubTab [1]). For comparisons, we also use 4 datasets from the OpenML-CC18 benchmark that were used for experiments in Section 4.3 of our main draft. As shown in the table below, all masking strategies show meaningful improvement over the no-masking case, where our marginal distribution masking shows the best result. In the revised draft (see Appendix H), we included the experimental details and results of the ablation study according to the types of masking.
>
> Note that marginal distribution masking is a popular masking (and augmentation) scheme in many tabular models, such as SubTab [1], VIME [2], and SCARF [3]. On the other hand, zero-masking and Gaussian noise may have higher chances of generating unrealistic data points. For example, zero-masking makes the data too sparse, and adding Gaussian noise to one-hot encoded categorical features is unrealistic.
>
> \begin{array}{lccccc}
> \hline
> \text{Masking type} & \text{income} & \text{cmc} & \text{semeion} & \text{pixel} & \text{Avg.} \newline
> \hline
> & \text{\\# shot = 1} & & & & \newline
> \hline
> \text{No-masking} & 59.18 & 36.23 & 54.04 & 75.99 & 56.36 \newline
> \text{Zero-masking} & 61.88 & 35.47 & 54.44 & 77.49 & 57.32 \newline
> \text{Gaussian noise} & 60.34 & 36.49 & 55.45 & 78.39 & 57.67 \newline
> \hline
> \textbf{Marginal distribution (Ours)} & \textbf{63.52} & \textbf{37.10} & \textbf{55.91} & \textbf{79.05} & \textbf{58.90} \newline
> \hline
> & \text{\\# shot = 5} & & & & \newline
> \hline
> \text{No-masking} & 70.94 & 38.76 & 72.12 & 86.38 & 67.05 \newline
> \text{Zero-masking} & 71.25 & 40.36 & 71.46 & 87.62 & 67.67 \newline
> \text{Gaussian noise} & 69.91 & 40.04 & \textbf{73.64} & 87.93 & 67.88 \newline
> \hline
> \textbf{Marginal distribution (Ours)} & \textbf{72.69} & \textbf{40.40} & 73.02 & \textbf{89.08} & \textbf{68.80} \newline
> \hline
> \end{array}
>
> [1] Ucar et al., “SubTab: Subsetting Features of Tabular Data for Self-Supervised Representation Learning”, NeurIPS 2021
>
> [2] Yoon et al., “VIME: Extending the Success of Self- and Semi-supervised Learning to Tabular Domain”, NeurIPS 2020
>
> [3] Bahri et al., “SCARF: Self-Supervised Contrastive Learning using Random Feature Corruption”, ICLR 2022

---

> ### Author Response · Authors · 2022-12-06
> **A Gentle Reminder**
>
> Dear reviewer 26Xs,
>
> Thank you very much again for your time and efforts in reviewing our paper.
>
> We kindly remind that we have only a week or so in the discussion period.
>
> We just wonder whether there is any further concern and hope to have a chance to respond before the discussion phase ends.
>
> Many thanks, Authors

---

> ### Author Response · Authors · 2022-12-13
> **Thank you for your response and careful reading on our rebuttal**
>
> Dear reviewer 26Xs,
>
> We are happy to hear that our response could help to address your concerns well.
>
> We appreciate not only your strong support on our paper, but also your careful reading on our draft and rebuttal (even those for other reviewers).
>
> Due to your valuable and constructive suggestions, we also believe that our paper is much improved.
>
> Thank you very much,\
> Authors

---

### Official Review · Reviewer_Gn88 · 2022-10-27

**Confidence:** 4
**Correctness:** 4
**Technical Novelty And Significance:** 3
**Empirical Novelty And Significance:** 4
**Recommendation:** 6

**Clarity, Quality, Novelty And Reproducibility:**

Clarity
The paper is well written and clear.

Novelty

The  work comes across as a minor improvement of the state of the art.
Reproducibility

The paper can certainly be reproduced because the work is described with sufficient clarity and code is provided.
Quality and Originality

The soundness of the results is commendable. The ideas are novel. I would like to be better convinced about the advance over the state of the art.

**Strength And Weaknesses:**

Strengths
1. Good motivation of the problem.
2. Thorough literature survey.
3. Sound adaptation of task generation based meta-learning to tabular data. Thorough understanding of the tabular data domain.
4. Good results

Weaknesses
1. Minor adaptation of the state of the art. Need to make clear why the proposed improvement is major.


**Summary Of The Paper:**

This paper proposes a meta-learning approach based on generation of new tasks from the columns of the training data. The authors's technique targets tabular data through modification of the task generation using randomization of the columns as well as an unsupervised validation technique. Through generation of such new tasks, the meta learning improves in it few shot learning performance. The proposed algorithm improves upon the state of the art notably.

**Summary Of The Review:**

The paper presents a technique that is well motivated by the needs of the target domain and addresses them well. The proposed technique achieves notable improvement over the state of the art. The advancement of the technique itself over the state of the art needs to be better established.

---

> ### Author Response · Authors · 2022-11-13
> **Response to Reviewer Gn88**
>
> Dear reviewer Gn88,
>
> We sincerely appreciate your efforts and insightful comments to improve the manuscript. We respond to each of your comments one-by-one in what follows. In the revised draft, we mark our revisions in “$\color{blue}{\text{blue}}$”.
>
> ----
>
> **[W1] Need to make clear why the proposed improvement is major: a minor improvement over the state-of-the-art?**
>
> The main reason why STUNT could advance state-of-the-art methods is that it is the only unsupervised meta-learning scheme specialized to tabular data. To be specific, it self-generates a diverse set of meta-training tasks from the unlabeled tabular data by treating the table's column feature as a useful target. Despite its simplicity, such an approach is well-suited due to the heterogeneous nature of the tabular dataset; one can consider any column features as a task label. Furthermore, we also propose an unsupervised validation scheme by utilizing STUNT to the unlabeled set for hyperparameter selection and early stopping. We indeed found that STUNT, the unsupervised meta-learning approach, is highly effective for few-shot semi-supervised tabular learning, compared to not only existing unsupervised meta-learning approaches, but also semi/self-supervised baselines.

---

> ### Author Response · Authors · 2022-12-06
> **A Gentle Reminder**
>
> Dear reviewer Gn88,
>
> Thank you very much again for your time and efforts in reviewing our paper.
>
> We kindly remind that we have only a week or so in the discussion period.
>
> We just wonder whether there is any further concern and hope to have a chance to respond before the discussion phase ends.
>
> Many thanks, Authors

---

### Author Response · Authors · 2022-11-13
**General response**

Dear reviewers and AC,

We sincerely appreciate your valuable time and effort spent reviewing our manuscript.

As reviewers highlighted, we believe our paper tackles an interesting and important problem (Gn88, 26Xs, sZ8N) and provides a novel (Gn88, sZ8N, 7dUE) and effective (26Xs, sZ8N, 7dUE) framework for few-shot tabular learning, validated with extensive evaluations (26Xs, 7dUE) followed by a clear presentation (Gn88, 7dUE).

We appreciate your constructive comments on our manuscript. In response to the comments, we have carefully revised and enhanced the manuscript with the following additional discussions and experiments:

- A clearer description of the primary goal, i.e., few-shot semi-supervised learning (in Abstract, Section 1).
- Ablation study (and analysis) of the masking design choice (in Appendix H).
- New baselines (and analysis): UMTRA, SES (in Section 2, Section 4.1, Appendix I, Table 1).
- References with additional discussion on self-supervised learning and unsupervised meta-learning (in Section 2).
- Experiment for few-shot tabular regression (in Appendix F).

These updates are temporarily highlighted in “$\color{blue}{\text{blue}}$” for your convenience to check.

We sincerely believe that STUNT can be a useful addition to the ICLR community, in particular, due to the above revision helping us better deliver the effectiveness of our method.

Thank you very much!

Authors.

---

### Decision · Program_Chairs · 2023-01-20

**Decision:**

Accept: notable-top-25%

**Justification For Why Not Higher Score:**

Four knowledgeable reviewers recommended acceptance, and one reviewer recommended rejection.  The authors addressed most of the reviewer's concerns, including answering clarifying questions, and providing additional results, including UMTRA and SES, and tabular regression tasks.

On balance, the AC recommends accepting the paper due to its contribution to few-shot learning knowledge for tabular data,  which is interesting to the ICLR audience.

**Justification For Why Not Lower Score:**

Four knowledgeable reviewers recommended acceptance, and one reviewer recommended rejection.  The authors addressed most of the reviewer's concerns, including answering clarifying questions, and providing additional results, including UMTRA and SES, and tabular regression tasks. On balance, the AC recommends accepting the paper due to its contribution to few-shot learning knowledge for tabular data,  which is interesting to the ICLR audience. The main drawback of this paper is that it is limited to tabular data, so it might have a limited influence on few-shot learning for vision or speech, for example.

**Metareview: Summary, Strengths And Weaknesses:**


The authors proposed a simple yet effective approach for few-shot learning on tabular data, dubbed Self-generated Tasks from UNlabeled Tables (STUNT). Their key idea is to treat randomly chosen columns as a target label to self-generate diverse few-shot tasks to learn a good representation.

Strengths
-------------
- Reviewer Gn88,Reviewer sZ8N : Good motivation of the problem,
- Reviewer Gn88: Thorough literature survey. Sound adaptation of task generation based meta-learning to tabular data.
- Reviewer Gn88, Reviewer 26Xs:  Thorough understanding of the tabular data domain. Limitations of the method are discussed with respect to low-shot learning.
- Reviewer Gn88, 26Xs, sZ8N,r 7dUE:  Good results. Experiments are fairly extensive and compare against a variety of baselines on several datasets and runs over multiple random seeds.
- Reviewer 26Xs, Reviewer 7dUE:: The proposed method is simple and intuitive.
-Reviewer sZ8N : The paper proposes a novel aspect of semi-supervised tabular data learning, which shows that unsupervised meta-learning is an effective tool for this task.
-Reviewer 7dUE: This paper is well-written and easy to follow. One can understand the proposed method by simply looking at figure 1.


Weaknesses
-------------

- Reviewer 26Xs: Novelty is somewhat low as the method could be viewed as a minor modification to CACTUs.
- Reviewer 26Xs: There are a few design decisions that are not well justified: the use of all columns for clustering in the pseudo-validation, the effect of marginal distribution masking versus other types of masking, and how the column masking ratio should be chosen.





**Note From Pc:**

if the above contains the word "oral" or "spotlight" please see: "oral" presentation means -> notable-top-5% and "spotlight" means -> notable-top-25%. As stated in our emails, we are disassociating presentation type from AC recommendations

**Summary Of Ac-Reviewer Meeting:**

NA